# Fair Online Bilateral Trade

**François Bachoc**
IMT
Université Paul Sabatier
Toulouse, France, 31062
Institut universitaire de France (IUF)
`francois.bachoc`
`@math.univ-toulouse.fr`

**Nicolò Cesa-Bianchi**
Department of Computer Science
Università degli Studi di Milano
DEIB
Politecnico di Milano
Milano, Italy, 20133
`cesa-bianchi@di.unimi.it`

**Tommaso Cesari**
EECS
University of Ottawa
Ottawa, Canada, K1N 6N5
`tcesari@uottawa.ca`

**Roberto Colomboni**
DEIB
Politecnico di Milano
Department of Computer Science
Università degli Studi di Milano
Milano, Italy, 20133
`roberto.colomboni@polimi.it`

## Abstract

In online bilateral trade, a platform posts prices to incoming pairs of buyers and sellers that have private valuations for a certain good. If the price is lower than the buyers' valuation and higher than the sellers' valuation, then a trade takes place. Previous work focused on the platform perspective, with the goal of setting prices maximizing the *gain from trade* (the sum of sellers' and buyers' utilities). Gain from trade is, however, potentially unfair to traders, as they may receive highly uneven shares of the total utility. In this work we enforce fairness by rewarding the platform with the *fair gain from trade*, defined as the minimum between sellers' and buyers' utilities. After showing that any no-regret learning algorithm designed to maximize the sum of the utilities may fail badly with fair gain from trade, we present our main contribution: a complete characterization of the regret regimes for fair gain from trade when, after each interaction, the platform only learns whether each trader accepted the current price. Specifically, we prove the following regret bounds: $\Theta(\ln T)$ in the deterministic setting, $\Omega(T)$ in the stochastic setting, and $\widetilde{\Theta}(T^{2/3})$ in the stochastic setting when sellers' and buyers' valuations are independent of each other. We conclude by providing tight regret bounds when, after each interaction, the platform is allowed to observe the true traders' valuations.

## 1 Introduction

In the online bilateral trade problem, at each round $t = 1, 2, \ldots$ a seller and a buyer with private valuations for a certain good connect to a trading platform. The seller's valuation $S_t$ is the smallest price at which they are willing to sell the good. Similarly, the buyer's valuation $B_t$ is the highest price they would pay to get the good. The platform posts a price $P_t$ to both buyer and seller. A trade happens if and only if both agents agree to trade, i.e., $S_t \le P_t \le B_t$. At the end of the round, $S_t$ and $B_t$ remain unknown and the platform only observes $\mathbb{I}\{S_t \le P_t\}$ (i.e., whether the seller accepted the deal) and $\mathbb{I}\{P_t \le B_t\}$ (i.e., whether the buyer accepted the deal).

Previous works [12, 2, 9, 13, 7, 5] focused on minimizing regret with respect to the gain from trade function GFT: $(p, s, b) \mapsto (b - s)\mathbb{I}\{s \le p \le b\}$. The quantity GFT$(P_t, S_t, B_t)$ corresponds to the sum

38th Conference on Neural Information Processing Systems (NeurIPS 2024).

|  | Determistic | Stochastic (i.i.d. + independent valuations) | Stochastic (i.i.d) |
|---|---|---|---|
| Two-bit feedback | $\ln T$ | $T^{2/3}$ | $T$ |
| Full feedback | 1 | $\sqrt{T}$ | $\sqrt{T}$ |

Table 1: Summary of our results.

of the seller's utility $P_t - S_t$ and the buyer's utility $B_t - P_t$ at time $t$ when a trade happens, and zero otherwise. This reward function, however, is oblivious to asymmetries in the utilities of the buyer and the seller caused by $P_t$ not being close to the mid-price $(S_t + B_t)/2$. As argued in [20], prices generating unequal gains may lead to a reduced participation in the market, which translates to less trading on the platform.

To address this problem, instead of using the *sum* of the utilities (i.e., the gain from trade) as reward function, we use the *minimum* of the utilities. This reward function ignores any surplus a trader may achieve at the expense of the other, thus encouraging the platform to set prices $P_t$ as close as possible to the mid-price, equalizing the profit of sellers and buyers. We call this new reward function *fair gain from trade* and denote it by FGFT. Our specific focus on the fair gain from trade objective can be seen as an implementation of the so-called egalitarian rule in social choice theory (sometimes also called the max-min rule or the Rawlsian rule), where one favors the alternative that maximizes the minimum utility of the involved parties to promote fairness. This approach might be especially relevant for online ride-sharing services like Uber and Lyft, where fairness problems have been previously studied, although in different settings with metrics different from ours (see, e.g., [31]). Our goal is to design algorithms minimizing the regret over $T$ rounds. This is the difference between the (expected) total FGFT achieved by the best fixed price $p^\star$ and the (expected) total FGFT achieved by the algorithm. Note that the two bits received as feedback at the end of each round (i.e., $\mathbb{I}\{S_t \le P_t\}$ and $\mathbb{I}\{P_t \le B_t\}$) are not enough to compute bandit feedback (the reward earned by posting the price $P_t$)—neither for FGFT nor for GFT.[1]

**Our contributions.** If sellers' and buyers' valuations are independent and drawn i.i.d. from two fixed but unknown distributions, we obtain an efficient algorithm (Algorithm 1), achieving a regret of $\widetilde{\mathcal{O}}(T^{2/3})$ after $T$ rounds (Theorem 2). This algorithm is built around the key *Convolution Lemma* (Lemma 1), which shows how one can estimate the *expected* FGFT through the feedback the learner has access to. Algorithm 1 does so by building uniform estimates for the expected FGFT via a discrete convolution procedure that combines the feedback collected from sellers and buyers across different time steps. We then derive a lower bound matching this rate up to a logarithmic factor (Theorem 3). The lower bound construction leverages the relationship between the feedback and the FGFT to build hard instances of our problem. These hard instances are similar to the ones in the *revealing action* problem of partial monitoring (see, e.g., [15]), that force the learner to perform a certain amount of costly exploration. An analogous phenomenon shows up even in the deterministic case, where $S_t = s$ and $B_t = b$ for all $t$ and for some unknown constants $s, b$. In this simpler setting, we prove that posting some clearly suboptimal but informative prices is unavoidable, showing that no strategy can obtain a regret better than $\Omega(\ln T)$ (Theorem 4). We complement this result by showing that this rate is matched by a double binary search algorithm (Theorem 5).

We also show that the independence of sellers' and buyers' valuations is necessary to minimize regret with respect to fair gain from trade: if the pairs $(S_t, B_t)$ are drawn i.i.d. from an arbitrary *joint* distribution, then the FGFT regret must grow linearly with time (Theorem 1).

Finally, we complete the picture by quantifying the cost of partial information. We do so by analyzing the regret rates at which the platform can learn if the pair $(S_t, B_t)$ is revealed at the end of each round—the so-called *full feedback* model. Here, we show that a regret of $\mathcal{O}(\sqrt{T})$ can be achieved for any *joint* distribution of sellers' and buyers' valuations (Theorem 6), and we show that this rate is optimal up to constant factors, even if we assume that the traders' valuations are independent of each other (Theorem 7).

Our results are summarized in Table 1.

---

[1]In fact, if a trade occurs, we only know that $S_t \le P_t \le B_t$, but this information alone allows us to compute neither $\min\{S_t - P_t, B_t - P_t\}$ nor $B_t - S_t$.

**Technical challenges.**    When regret is minimized with respect to the *gain from trade*, the independence of sellers' and buyers' valuations is not enough to guarantee sublinear regret, and an additional *smoothness* assumption[2] is required to compensate for the lack of Lipschitzness of GFT. In [9, 10] the optimal regret for GFT under the smoothness assumption turns out to be the same as the optimal rate $\widetilde{\Theta}(T^{2/3})$ achievable for FGFT without smoothness. This happens because the fairness condition confers Lipschitzness to the gain from trade, allowing us to compare against a broader range of distributions. In fact, the smoothness assumption plays no role in the fair version of the problem. To further compare the two settings, note that the issue appearing in [9], i.e., a feedback so poor that is not even sufficient to reconstruct the reward at the posted price, arises here as well. In our case, however, the specific form of the objective requires new ideas—i.e., our new Convolution Lemma (Lemma 1)—to recover usable information about the reward function. Another difference is that, in [9], the realized gain from trade is maximized by posting any price $p \in [S, B]$. In our setting, instead, we have to address the more delicate task of locating the midpoint $p = (S + B)/2$. This is the reason why, as we show in Section 2, even optimal algorithms for bilateral trade can suffer linear regret in our setting. From the technical viewpoint, note that a direct application of the convolution lemma in the 2-bit feedback setting would yield a suboptimal upper bound of $T^{3/4}$. To obtain the optimal $T^{2/3}$ rate, we carefully define a data-gathering procedure in which each observation contributes to estimating the convolution of the cdfs at all points.

Another interesting discrepancy between the two settings arises in the deterministic case, where a learner can devise a strategy whose regret is constant when the reward function is GFT. This is in contrast to the $\Theta(\ln T)$ rate when the reward function is FGFT, as we discuss at the beginning of Section 4.

**Related work.**    Attempts to circumvent the linear lower bound for the regret of gain from trade in adversarial environments include [2], where they focus on 2-regret, and [5], where they consider a global budget balance condition that allows the learner to subsidize trades with money accumulated in previous rounds. Recently, online brokerage, a related though incomparable setting where traders can sell or buy depending on the contingent market conditions, has been studied in [7, 17, 3].

Fairness is an intensively studied topic in online learning, with the goal of understanding the extent to which the fairness constraints impact on the regret. The work [6] considers online prediction with expert advice and studies the problem of combining individually non-discriminatory experts while preserving non-discrimination. An early investigation of fairness in linear bandits is conducted in [23], where the fairness constraints demand that similar action be assigned approximately equal probabilities of being pulled, and the similarity metric must be learned via fairness violation feedback. Fairness in linear bandits is also investigated in [22], where the reward observed by the learner is biased towards a specific group of actions. In [4], the authors study an online binary classification problem with one-sided feedback where the fairness constraint requires the false positive rate to be equal across two groups of incoming users. The paper [19] applies Blackwell's approachability theory to investigate online learning under group fairness constraints. A different notion of fairness in bandits is considered by [28, 30], where each arm has to be pulled at least a pre-specified fraction of times (see also [8] for related results). This type of fairness requirement is also considered in [32] where the optimal policy is pulling an arm with a probability proportional to its merit (a problem-specific function of the arm's expected reward). Finally, the work [24] considers a $K$-armed bandit setting in which pulling an arm yields different rewards for different agents. The algorithm's goal is to control regret against the optimal Nash Social Welfare (NSW). The NSW of a probability assignment $\pi$ over the arms is the product of the agents' reward in expectation according to $\pi$.

Online learning with fairness constraints is also investigated in online fair division [1], where each good in a sequence must be allocated to a set of agents who receive some utility that depends on the good. The goal is to satisfy a given fairness criterion, which is typically not aligned with the maximization of the agents' utilities. Traditionally, this problem has been studied under the assumption that at the beginning of each round agents report their true utilities [29]. Very recently, the problem was studied in a bandit setting, where only the stochastic utility of the agent receiving the good is revealed [33]. In that setting, there is a finite set of types and the expected utility of agent $i$ for type $j$ is fixed but unknown. The regret is then defined in terms of the geometric mean of the total expected utilities of the agents. Our regret, instead, is defined additively over the minimum

---

[2]A distribution is smooth whenever it admits a bounded density; see, e.g., [21, 11].

of the agents' utilities in each round, where the utilities of sellers and buyers depend on each other through the price posted by the platform. Our notion of regret can also be viewed as an online version of the Kalai–Smorodinsky solution to the bargaining problem, in the sense that we also strive to equalize the utilities of the two players [25]. The regret typically studied in online fair division, instead, corresponds to the Nash solution to the bargaining problem.

Finally, we mention that the line of research where online learning is applied to trading problems, was initiated in one-sided settings, like dynamic pricing, whose seminal paper was [26]. For an interesting comparison of (a generalization of) the dynamic pricing setting and the bilateral trade setting, see also [14].

**Formal problem definition.** We study the following problem. At each time $t \in \mathbb{N}$,

1. A seller and a buyer arrive with private valuations $S_t \in [0,1]$ and $B_t \in [0,1]$
2. The platform proposes a trading price $P_t \in [0,1]$
3. If $S_t \leq P_t \leq B_t$, then the buyer gets the object and pays $P_t$ to the seller
4. $\mathbb{I}\{S_t \leq P_t\}$ and $\mathbb{I}\{P_t \leq B_t\}$ are revealed

The boundedness assumption for valuations and prices is standard in regret minimization settings. We enforce fairness by rewarding the platform with the minimum of the utilities of sellers and buyers. More precisely, for any $p \in [0,1]$ and any $s, b \in [0,1]$, we define the *fair gain from trade* achieved with $p$ when the seller's valuation is $s$ and the buyer's valuation is $b$ by

$$\textsc{fgft}(p, s, b) \coloneqq \min\left\{(p-s)_+, (b-p)_+\right\},$$

where $x_+ \coloneqq \max\{x, 0\}$ for any $x \in \mathbb{R}$.

We assume a stochastic model where the sequence $(S_t, B_t)_{t \in \mathbb{N}}$ of sellers' and buyers' valuations is an i.i.d. process with a fixed but unknown distribution. The regret after $T$ rounds of an algorithm posting prices $P_1, P_2, \ldots$ is defined by

$$R_T \coloneqq \max_{0 \leq p \leq 1} \mathbb{E}\left[\sum_{t=1}^{T} \textsc{fgft}(p, S_t, B_t)\right] - \mathbb{E}\left[\sum_{t=1}^{T} \textsc{fgft}(P_t, S_t, B_t)\right].$$

In the deterministic setting (which can be viewed as a special case of the above stochastic setting) there exist $s, b \in [0,1]$ such that, for every $t \in \mathbb{N}$, it holds that $S_t = s$ and $B_t = b$. In both cases, note that the maximum exists because the expected FGFT is a 1-Lipschitz function of the price.

## 2 Maximization of gain from trade does not imply fairness

In this section, we show that, in general, a no-regret algorithm for GFT fails to achieve no-regret guarantees for FGFT.

Consider the stochastic setting, where the sequence $(S_t, B_t)_{t \in \mathbb{N}}$ is an i.i.d. process. Pick $0 < h < 1/2$ and, for all $t \in \mathbb{N}$, let $S_t$ be such that $S_t = 0$ with probability $1/2$ and $S_t = 1 - h$ with probability $1/2$. Let also $B_t = 1$ for all $t \in \mathbb{N}$. Then, the only prices maximizing GFT are those in the interval $[1-h, 1]$. Now, any price $p \in [1-h, 1]$ achieves an expected FGFT of

$$\mathbb{E}\left[\textsc{fgft}(p, S_t, B_t)\right] = \frac{1}{2} \cdot (1-p) + \frac{1}{2} \cdot \min\left\{(1-p), \left(p - (1-h)\right)\right\}$$

and the maximum of this quantity is $h/2$, which is attained, for example, by posting $p = 1 - h$. On the other hand, it is easy to see that the maximum of the expected FGFT on the whole interval $[0,1]$ is $1/4 > h/2$, achieved by posting $p = 1/2 \notin [1-h, 1]$. So, if we use any no-regret algorithm for the standard bilateral trade problem to post prices $P_1, P_2, \ldots$, we suffer linear FGFT regret on this instance. Even more strikingly, on these instances, for any $t \in \mathbb{N}$,

$$\frac{\mathbb{E}\left[\textsc{fgft}(1-h, S_t, B_t)\right]}{\mathbb{E}\left[\textsc{fgft}(1/2, S_t, B_t)\right]} = 2h \to 0^+ \qquad \text{for } h \to 0^+,$$

which implies that there are instances where any no-regret algorithm for the standard bilateral trade problem fails even if we content ourselves with competing against a fraction of the reward earned by a no-regret algorithm for the fair bilateral trade problem.[3]

## 3 The stochastic case

We begin by providing a linear lower bound on the worst-case regret for the stochastic case. The idea of the proof is to leverage a *lack of observability* phenomenon: we can devise two different distributions whose maximum expected fair gain from trade is achieved in disjoint regions—so that posting a price that is in the good region for one distribution leads to an instantaneous regret bounded away from zero for the other distribution—but the learner cannot distinguish which is the underlying distribution generating the valuations, given that these distributions are designed so that the (push-forward) distribution of the received feedback is exactly the same for both of them.

**Theorem 1.** *In the stochastic case, for every algorithm for the fair bilateral trade problem, there exists a joint distribution under which, for an i.i.d. sequence $(S_t, B_t)_{t \in \mathbb{N}}$ of sellers and buyers, we have, for all $T \in \mathbb{N}$,*

$$R_T \geq \frac{T}{48} \ .$$

*Proof.* For any point $x \in [0,1]^2$, let $\delta_x$ be the Dirac measure centered at $x$. Consider the two distributions $\mu \coloneqq \frac{1}{3}\left(\delta_{(0, \frac{5}{8})} + \delta_{(\frac{3}{8}, \frac{3}{8})} + \delta_{(\frac{5}{8}, 1)}\right)$ and $\nu \coloneqq \frac{1}{3}\left(\delta_{(0, \frac{3}{8})} + \delta_{(\frac{3}{8}, 1)} + \delta_{(\frac{5}{8}, \frac{5}{8})}\right)$. Suppose that the sequence of valuations $(S_t, B_t)_{t \in \mathbb{N}}$ is drawn i.i.d. from $\mu$ or $\nu$. If the underlying distribution is $\mu$, the optimal point is $5/16$ and, for all $p \in [1/2, 1]$ and $t \in \mathbb{N}$, the difference $\mathbb{E}\left[\textsc{fgft}(5/16, S_t, B_t)\right] - \mathbb{E}\left[\textsc{fgft}(p, S_t, B_t)\right]$ is at least $\frac{1}{3}\left(\frac{5}{16} - \frac{3}{16}\right) = \frac{1}{24}$. Analogously, if the underlying distribution is $\nu$, the optimal point is $11/16$ and, for all $p \in [0, 1/2]$ and $t \in \mathbb{N}$, $\mathbb{E}\left[\textsc{fgft}(11/16, S_t, B_t)\right] - \mathbb{E}\left[\textsc{fgft}(p, S_t, B_t)\right] \geq \frac{1}{24}$. This means that the only way for the learner not to suffer $\Omega(T)$ regret is to distinguish whether the underlying distribution is $\mu$ or $\nu$. But a direct verification shows that the distribution of the feedback is the same for every $p \in [0, 1]$, regardless of whether the underlying distribution is $\mu$ or $\nu$. Hence, the learner has no means to distinguish between $\mu$ and $\nu$ and must suffer $\Omega(T)$ regret. Indeed, let $N_T$ be the random number of times the platform posts a price $P_t$ in $[0, \frac{1}{2}]$. Then $N_T$ has the same distribution under $\mu$ and $\nu$. If $\mathbb{E}_\mu[N_T] = \mathbb{E}_\nu[N_T] \geq \frac{T}{2}$, then the expected regret under $\nu$ is at least $\frac{T}{2} \cdot \frac{1}{24} = \frac{T}{48}$. Conversely, If $\mathbb{E}_\mu[N_T] = \mathbb{E}_\nu[N_T] \leq \frac{T}{2}$, then the expected regret under $\mu$ is at least $\frac{T}{48}$. $\qquad\square$

**Remark 1.** *Note that Theorem 1 together with Yao's Minimax Theorem immediately imply that the* adversarial *fair bilateral trade problem—where the goal is to obtain sublinear worst-case expected regret against the best fixed-price when the sequence of seller/buyer valuations $(S_t, B_t)_{t \in \mathbb{N}}$ is chosen by an oblivious adversary—is unlearnable.*

We now show that we can achieve learnability in the stochastic case by assuming that, for each $t \in \mathbb{N}$, the two valuations $S_t$ and $B_t$ are *independent* of each other.

To this end, we first present the Convolution Lemma (Lemma 1), which provides a way to avoid the aforementioned lack of observability when the traders' valuations are independent of each other. This lemma plays for FGFT a role analogous to the one played by the Decomposition Lemma [9, Lemma 1] for GFT.

**Lemma 1** (The Convolution Lemma)**.** *For all $s, b, p \in [0,1]$,*

$$\textsc{fgft}(p, s, b) = \int_0^1 \mathbb{I}\{s \leq p - u\}\mathbb{I}\{p + u \leq b\}\, \mathrm{d}u \ . \tag{1}$$

*In particular, if $S$ and $B$ are $[0,1]$-valued independent random variables, for each $p \in [0,1]$,*

$$\mathbb{E}\left[\textsc{fgft}(p, S, B)\right] = \int_0^1 \mathbb{P}[S \leq p - u]\mathbb{P}[p + u \leq B]\, \mathrm{d}u \ . \tag{2}$$

---

[3]The reader familiar with the notion of $\alpha$-regret will note that this is equivalent to saying that any no-regret algorithm for the standard bilateral trade problem fails to be a no-$\alpha$-regret algorithm for the *fair* bilateral trade problem, regardless of how large $\alpha$ is chosen.

*Proof.* If $\text{FGFT}(p, s, b) = (b - p)_+$, note that

$$(b - p)_+ = \int_0^1 \mathbb{I}\{u \le (b - p)_+\} \, \mathrm{d}u = \int_0^1 \mathbb{I}\{u \le (p - s)_+\} \mathbb{I}\{u \le (b - p)_+\} \, \mathrm{d}u$$

$$= \int_0^1 \mathbb{I}\{u \le p - s\} \mathbb{I}\{u \le b - p\} \, \mathrm{d}u = \int_0^1 \mathbb{I}\{s \le p - u\} \mathbb{I}\{p + u \le b\} \, \mathrm{d}u \ .$$

The same conclusion holds with an analogous argument if $\text{FGFT}(p, s, b) = (p - s)_+$. The stochastic case follows immediately from Fubini's theorem and the independence of $S$ and $B$. $\qquad\square$

We now explain how the previous lemma can be used to recover the observability of the expected FGFT under the independence assumption. Note that, given the feedback we have access to, we can estimate the cumulative distributions of both sellers' and buyers' valuations pointwise with arbitrary precision, and hence, *a fortiori*, we can estimate (2). However, even if this observation alone would be enough to ensure learnability, this is not the most efficient way of learning the expected fair gain from trade function. In fact, we do not really need a careful pointwise estimation of both cumulative distributions, but just of (2), which involves certain products of their translates. The crucial observation—from which Lemma 1 takes its name—is that (2) is the (incomplete) *convolution* of the cumulative distribution of the sellers' valuations and the co-cumulative distribution of the buyers' valuations, and that we have noisy access to these functions at the points we need to estimate them, though at different time steps. This observation suggests that we can approximate the continuous (incomplete) convolution by a discrete (incomplete) convolution involving the noisy observations we collect at different time steps, e.g., by posting prices on a uniform grid. These ideas are exploited in the design of Algorithm 1 and in the proof of its regret guarantees.

---

**Algorithm 1:** Convolution Pricing (Stochastic Setting)

---

**Input**: $K \in \mathbb{N}$;
**Initialization**: for each $t \in \mathbb{Z}$, set $V_t \coloneqq W_t \coloneqq 0$;
**for** *time* $t = 1, 2, \ldots, K$ **do**
$\qquad$ Post price $P_t \coloneqq \frac{t}{K}$ and set $V_t \coloneqq \mathbb{I}\{S_t \le P_t\}, W_t \coloneqq \mathbb{I}\{P_t \le B_t\}$;
Let $I \in \text{argmax}_{i \in [K]} \frac{1}{K} \sum_{k=0}^{K-1} V_{i-k} W_{i+k}$;
**for** $t = K + 1, K + 2, \ldots, T$ **do**
$\qquad$ Post price $P_t \coloneqq I/K$;

---

**Theorem 2.** *In the stochastic case, under the additional assumption that for each $t \in \mathbb{N}$ the seller's valuation $S_t$ is independent of the buyer's valuation $B_t$, by setting $K \coloneqq \lfloor T^{2/3} \rfloor$, the regret suffered by Algorithm 1 is $\widetilde{O}(T^{2/3})$.*

*Proof.* For each $k \in [K]$, define $q_k \coloneqq \frac{k}{K}$. Note that for each $t \in \mathbb{N}$ the function $p \mapsto \mathbb{E}[\text{FGFT}(p, S_t, B_t)]$ is 1-Lipschitz being the expectation of (random) 1-Lipschitz functions. Hence, if for each $p \in [0, 1]$ we denote by $k^\star(p)$ the index of the closest point to $p$ in the grid $\{q_1, \ldots, q_K\}$, we have, for each $t \in \mathbb{N}$, that,

$$\mathbb{E}\big[\text{FGFT}(p, S_t, B_t)\big] - \mathbb{E}\big[\text{FGFT}\big(q_{k^\star(p)}, S_t, B_t\big)\big] \le \frac{1}{K} \ . \tag{3}$$

Let $F$ be the common cumulative function of the random variables in the process $(S_t)_{t \in \mathbb{N}}$ and let $G$ be the common co-cumulative function of the random variables in the process $(B_t)_{t \in \mathbb{N}}$, i.e., for each $t \in \mathbb{N}$ and each $p \in \mathbb{R}$, define $F(p) \coloneqq \mathbb{P}[S \le p]$ and $G(p) \coloneqq \mathbb{P}[p \le B]$. Note that $F(u) = 0$ for each $u \le 0$, that $G(u) = 0$ for each $u \ge 1$, and that for each $p \in [0, 1]$, the function $u \mapsto F(p - u)G(p + u)$ is non-increasing, being the product of two non-increasing functions. Hence, for each $t \in \mathbb{N}$ and each

$k \in [K]$, by Lemma 1, we can sandwich the quantity $\frac{1}{K} \sum_{i=0}^{K-1} F(q_{k-i}) G(q_{k+i})$ as follows:

$$\mathbb{E}\big[\text{FGFT}(q_k, S_t, B_t)\big] = \int_0^1 F(q_k - u)G(q_k + u)\,\mathrm{d}u = \sum_{i=0}^{K-1} \int_{\frac{i}{K}}^{\frac{i+1}{K}} F(q_k - u)G(q_k + u)\,\mathrm{d}u$$

$$\leq \sum_{i=0}^{K-1} \int_{\frac{i}{K}}^{\frac{i+1}{K}} F\left(\frac{k}{K} - \frac{i}{K}\right)G\left(\frac{k}{K} + \frac{i}{K}\right)\mathrm{d}u = \frac{1}{K}\sum_{i=0}^{K-1} F(q_{k-i})G(q_{k+i})$$

$$\leq \frac{1}{K} + \frac{1}{K}\sum_{i=0}^{K-1} F(q_{k-(i+1)})G(q_{k+(i+1)}) = \frac{1}{K} + \sum_{i=0}^{K-1}\int_{\frac{i}{K}}^{\frac{i+1}{K}} F\left(\frac{k}{K} - \frac{i+1}{K}\right)G\left(\frac{k}{K} + \frac{i+1}{K}\right)\mathrm{d}u$$

$$\leq \frac{1}{K} + \sum_{i=0}^{K-1}\int_{\frac{i}{K}}^{\frac{i+1}{K}} F(q_k - u)G(q_k + u)\,\mathrm{d}u = \frac{1}{K} + \int_0^1 F(q_k - u)G(q_k + u)\,\mathrm{d}u$$

$$= \frac{1}{K} + \mathbb{E}\big[\text{FGFT}(q_k, S_t, B_t)\big]. \tag{4}$$

Now, by the independence assumption,

$$\frac{1}{K}\sum_{i=0}^{K-1} F(q_{k-i})G(q_{k+i}) = \frac{1}{K}\sum_{i=0}^{K-1} \mathbb{E}[V_{k-i}]\mathbb{E}[W_{k+i}] = \mathbb{E}\left[\frac{1}{K}\sum_{i=0}^{K-1} V_{k-i}W_{k+i}\right]$$

and, by noting that for each $k \in [K]$ we have that $\frac{1}{K}\sum_{i=0}^{K-1} V_{k-i}W_{k+i}$ is the empirical mean of $K$ $\{0,1\}$-valued independent random variables, by Hoeffding's inequality and a union bound, we have that, for any $\varepsilon > 0$,

$$\mathbb{P}\left[\max_{k\in[K]} \left|\frac{1}{K}\sum_{i=0}^{K-1} V_{k-i}W_{k+i} - \frac{1}{K}\sum_{i=0}^{K-1} F(q_{k-i})G(q_{k+i})\right| \geq \varepsilon\right] \leq 2K\exp(-2\varepsilon^2 K).$$

In particular, if we set $\varepsilon_T \coloneqq \sqrt{\frac{\ln(2T)}{2\lfloor T^{2/3}\rfloor}}$, recalling that $K = \lfloor T^{2/3}\rfloor$, and defining the (good) event $\mathcal{E}_T \coloneqq \{\max_{k\in[K]} |\frac{1}{K}\sum_{i=0}^{K-1} V_{k-i}W_{k+i} - \frac{1}{K}\sum_{i=0}^{K-1} F(q_{k-i})G(q_{k+i})| < \varepsilon_T\}$, we have

$$\mathbb{P}[\mathcal{E}_T^c] \leq \frac{1}{T^{1/3}}. \tag{5}$$

Now, let $p^\star \in \text{argmax}_{p\in[0,1]} \mathbb{E}\big[\text{FGFT}(p, S_t, B_t)\big]$ (whose definition is independent of $t$, given that the process $(S_t, B_t)_{t\in\mathbb{N}}$ is i.i.d.) and, recalling the definition of $I$ from Algorithm 1, note, for each $t > K$, that

$$\mathbb{E}\big[\text{FGFT}(P_t, S_t, B_t)\big] = \mathbb{E}\big[\text{FGFT}(q_I, S_t, B_t)\big] = \mathbb{E}\big[\mathbb{E}\big[\text{FGFT}(q_I, S_t, B_t) \mid I\big]\big]$$

$$= \mathbb{E}\left[\big[\mathbb{E}\big[\text{FGFT}(q_k, S_t, B_t)\big]\big]_{k=I}\right] \geq \mathbb{E}\left[\left[\frac{1}{K}\sum_{i=0}^{K-1} F(q_{k-i})G(q_{k+i}) - \frac{1}{K}\right]_{k=I}\right]$$

$$= \mathbb{E}\left[\frac{1}{K}\sum_{i=0}^{K-1} F(q_{I-i})G(q_{I+i})\right] - \frac{1}{K} \geq \mathbb{E}\left[\mathbb{I}_{\mathcal{E}_T}\cdot\frac{1}{K}\sum_{i=0}^{K-1} V_{I-i}W_{I+i}\right] - \varepsilon_T - \frac{1}{K}$$

$$\geq \mathbb{E}\left[\mathbb{I}_{\mathcal{E}_T}\cdot\frac{1}{K}\sum_{i=0}^{K-1} V_{k^\star(p^\star)-i}W_{k^\star(p^\star)+i}\right] - \varepsilon_T - \frac{1}{K} \geq \mathbb{E}\left[\frac{1}{K}\sum_{i=0}^{K-1} V_{k^\star(p^\star)-i}W_{k^\star(p^\star)+i}\right] - \mathbb{P}[\mathcal{E}_T^c] - \varepsilon_T - \frac{1}{K}$$

$$= \frac{1}{K}\sum_{i=0}^{K-1} F(q_{k^\star(p^\star)-i})G(q_{k^\star(p^\star)+i}) - \mathbb{P}[\mathcal{E}_T^c] - \varepsilon_T - \frac{1}{K} \geq \mathbb{E}\big[\text{FGFT}(p^\star, S_t, B_t)\big] - \mathbb{P}[\mathcal{E}_T^c] - \varepsilon_T - \frac{2}{K},$$

where the third equality follows from the Freezing Lemma (see, e.g., [18, Lemma 8]), the first inequality from the sandwich inequalities in (4), the second inequality from the definition of $\mathcal{E}_T$, the third inequality from the definition of $I$, and the last inequality from the sandwich inequalities in (4) and inequality (3). Putting everything together, we can upper bound the regret as follows

$$R_T \leq K + \sum_{t=K+1}^T \mathbb{E}\big[\text{FGFT}(p^\star, S_t, B_t) - \text{FGFT}(P_t, S_t, B_t)\big] \leq K + \sum_{t=K+1}^T \left(\mathbb{P}[\mathcal{E}_T^c] + \varepsilon_T + \frac{2}{K}\right).$$

Recalling that $K = \lfloor T^{2/3}\rfloor$, $\varepsilon_T = \sqrt{\frac{\ln(2T)}{2\lfloor T^{2/3}\rfloor}}$, and (5), we obtain the conclusion. □

We now prove that the strategy employed by Algorithm 1 is worst-case optimal, up to logarithmic factors. At a high level, the reason why the $T^{2/3}$ rate is optimal is that the fair bilateral trade problem contains instances that closely resemble the revealing action problem in partial monitoring [15], where, in order to distinguish which one of two actions is optimal, we have to play for a significant amount of time a third highly suboptimal action to gather this information. We formalize this intuition in the following theorem, whose full proof is deferred to Appendix A due to space constraints.

**Theorem 3.** *There exists a constant $c > 0$ such that the following holds. For every algorithm for the fair bilateral trade problem and for every time horizon $T \in \mathbb{N}$, there exists an i.i.d. sequence $(S_t, B_t)_{t \in [T]}$ of pairs of sellers' and buyers' valuations such that, for each $t \in [T]$, $S_t$ is independent of $B_t$, and the algorithm suffers regret of at least $cT^{2/3}$ on that sequence.*

*Proof sketch.* For each $\varepsilon \in [-1, 1]$, consider the distribution $\mu_\varepsilon := \frac{1+\varepsilon}{2}\delta_0 + \frac{1-\varepsilon}{2}\delta_{1/4}$ where $\delta_a$ is the Dirac measure centered at a point $a \in \mathbb{R}$. For each $\varepsilon \in [-1, 1]$, let $(S_t^\varepsilon)_{t \in \mathbb{N}}$ be an i.i.d. sequence whose distribution is $\mu_\varepsilon$. For each $t \in \mathbb{N}$, let $B_t = 1$. Then, for each $\varepsilon \in [-1, 1]$, the sequence $(S_t^\varepsilon, B_t)_{t \in \mathbb{N}}$ is i.i.d. and for each $t \in \mathbb{N}$ the valuation $S_t^\varepsilon$ is (obviously) independent of the (deterministic) valuation $B_t$. Now, one can show that if $\varepsilon > 0$ the optimal price to post in order to maximize the expected fair gain from trade over the sequence $(S_t^\varepsilon, B_t)_{t \in \mathbb{N}}$ is $1/2$ while if $\varepsilon < 0$ the optimal price is $5/8$. If $\varepsilon > 0$ and we post prices greater or equal than $9/16$ we suffer instantaneous regret of at least order of $|\varepsilon|$, while if $\varepsilon < 0$ and we post prices less or equal than $9/16$ we suffer instantaneous regret of at least order of $|\varepsilon|$. Hence, to avoid suffering $\Omega(|\varepsilon|T)$ regret we need to distinguish the sign of $\varepsilon$. Given that what we see are the two bits in the feedback, the only way to discriminate the sign of $\varepsilon$ is to post prices in the region between $0$ and $1/4$. In this case, the feedback we see is equivalent to seeing a Bernoulli of parameter $\frac{1+\varepsilon}{2}$, and hence, due to information-theoretic arguments, we need $\Omega(1/\varepsilon^2)$ to distinguish the sign of $\varepsilon$. Now, every price in the region $[0, 1/4]$ suffers instantaneous regret of order $\Omega(1)$. Hence, any algorithm has to suffer $\Omega\left(\frac{1}{\varepsilon^2} + |\varepsilon|T\right)$ regret, which is $\Omega(T^{2/3})$ when $|\varepsilon| = \Theta(T^{-1/3})$. □

**Remark 2.** *The $T^{2/3}$ rate we achieve in Theorem 3 was also the regret rate for the standard bilateral trade problem in [9], where in order to achieve learnability, on top of the stochastic and independent valuations assumptions, it was also required that the valuations admitted a bounded density. In that case, the regret rate degraded multiplicatively with the upper bound on that density. Instead, we manage to obtain this rate without the extra bounded density assumption and our bound does not explode if the bounded density constant diverges. The reason is that, differently from the discontinuous gain from trade function, the fair gain from trade is $1$-Lipschitz.*

## 4 The deterministic case

In this section, we study the deterministic case where there exist fixed but unknown constants $s, b \in [0, 1]$ such that, for all time $t \in \mathbb{N}$, $S_t = s$ and $B_t = b$. In this case, we note that if $b < s$ no trade can occur while, if $b = s$, even though a trade can occur for the posted price $P_t = b = s$, no gain from trade or fair gain from trade can be obtained from it. Consequently, we focus on the only interesting case $s < b$.

We begin by the following remark: in the standard bilateral trade setting where the reward function is the gain from trade $p \mapsto (b - s)\mathbb{I}\{s \le p \le b\}$, we can devise an algorithm that achieves *constant* regret. In fact, the learner can post prices following a binary search (starting by posting the price $1/2$), move to the next dyadic point to the left (resp., right) if both valuations were lower (resp., higher) than the proposed price, while keep playing the same price as soon as a successful (dyadic) price $p$ is proposed, i.e., a price $p \in [s, b]$. This way, if the learner fails $n$ times before the first success, an upper bound on the cumulative regret suffered in the "failure" phase is $n \cdot 2^{-n} \le 1$, and from the subsequent rounds the instantaneous regret is always zero.

In contrast, the deterministic fair bilateral trade problem is still sufficiently layered that a costly exploration phase is unavoidable to achieve learnability given the intertwined relationship between the reward function and the feedback. This is also in contrast to what happens in the stochastic setting, where the standard bilateral trade problem required the extra bounded density assumption to achieve learnability (and hence, in this sense, was harder) than the fair bilateral trade problem.

**Theorem 4.** *In the deterministic case, for any horizon $T \geq 17$, any algorithm suffers a worst-case regret larger than or equal to $\frac{1}{32}\log_2(T)$.*

For the full proof of this result, see Appendix B.

*Proof sketch.* Since the setting is deterministic, we can restrict the proof to deterministic algorithms without loss of generality. Then, the key property we leverage is that for any $k \in \mathbb{N}$, if a deterministic algorithm posts at most $k$ prices in $\left[0, \frac{1}{4}\right]$, then there is an interval $E_k \subset \left[0, \frac{1}{4}\right]$ of length $\Theta(2^{-k})$ such that, for each $s, s' \in E_k$, the algorithm receives the same feedback from the environments defined by seller/buyer pairs $(s, 1)$ and $(s', 1)$, therefore selecting the same prices.

Now, fix a deterministic algorithm $\alpha$. There exists $k^\star \in \{0, 1, \ldots, T\}$ such that, for all $s \in E_{k^\star}$, the algorithm posts exactly $k^\star$ prices in $\left[0, \frac{1}{4}\right]$. A direct verification shows that, for any $s \in E_{k^\star}$, the instantaneous regret paid for the time steps where $P_t \in \left[0, \frac{1}{4}\right]$ is $\Omega(1)$. Therefore, the regret paid by the algorithm for playing $k^\star$ times in $\left[0, \frac{1}{4}\right]$ is $\Omega(k^\star)$. Moreover, let $\mathcal{T}$ be the set of time steps $t \in [T]$ where the algorithm posts prices $P_t \in \left(\frac{1}{4}, 1\right]$. Leveraging the fact that the algorithm posts the same prices for every $s \in E_{k^\star}$, it can be proved that there exists an $s \in E_{k^\star}$ such that the average distance $\frac{1}{|\mathcal{T}|}\sum_{t \in \mathcal{T}}\left|P_t - \frac{b+s}{2}\right|$ of the points $P_t$ played at rounds $t \in \mathcal{T}$ from the maximizer $\frac{b+s}{2}$ of the FGFT is at least $\Omega(2^{-k^\star})$. Putting everything together, the worst-case regret of the algorithm is lower bounded by $\Omega\left(k^\star + (T - k^\star)2^{-k^\star}\right)$, which is $\Omega(\ln T)$ regardless of the specific value of $k^\star$. $\qquad\square$

A "double" binary search algorithm suffices to obtain a matching $O(\ln T)$ regret rate. The idea of the algorithm is very simple. First, spend $\Theta(\ln T)$ rounds performing a binary search for the valuation of the seller. Then, do the same for the valuation of the buyer. Finally, commit to the average of these two estimates for the remaining time steps. For completeness, we report the pseudocode of this algorithm in Appendix C (Algorithm 3). For a full proof of the following theorem, see Appendix C.

**Theorem 5.** *In the deterministic case, the (deterministic) regret suffered by Algorithm 3 is $O(\ln T)$.*

*Proof sketch.* The idea is first to invest a budget of order $\ln T$ to locate the seller's valuation with error $O(1/T)$. Second, we proceed similarly to locate the buyer's valuation with error $O(1/T)$. Finally, this yields a $O(1/T)$-precise estimate of the optimal price $(s + b)/2$, and we commit to this estimate for the remaining time steps. The regret incurred by the two first phases is of order $\ln T$ while the regret of the last phase is of order $T \cdot \frac{1}{T}$, and thus bounded. $\qquad\square$

## 5 The full-feedback model

We conclude this paper by quantifying the cost of partial information by analyzing the full feedback model, where, after posting the price $P_t$, the learner has access to both $S_t$ and $B_t$. We show that the cost is two-fold: slower rates (both in the deterministic —$\Theta(\ln T)$ vs $\Theta(1)$— and stochastic case —$\Theta(T^{2/3})$ vs $\Theta(\sqrt{T})$) and the need for additional assumptions (independence of buyer and seller valuations in the stochastic case).

For the full-feedback setting, we show that following the best empirical price leads to an algorithm (Algorithm 2) whose regret guarantees are optimal. We remark that Algorithm 2 needs full feedback to run, given that it needs to compute the FGFT function for counterfactual prices. For the full proof of the next theorem, see Appendix D.

---

**Algorithm 2:** Follow the Best Empirical Price

---

**Initialization**: Select $P_1 := 1/2$;
**for** *time $t = 1, 2, \ldots$* **do**
    Post price $P_t$, and receive feedback $(S_t, B_t)$;
    Select $P_{t+1} \in \arg\max_{p \in [0,1]} \frac{1}{t}\sum_{s=1}^{t} \text{FGFT}(p, S_t, B_t)$

---

**Theorem 6.** *In the stochastic full-feedback case, the regret suffered by Algorithm 2 is $O(\sqrt{T})$. In the deterministic setting, the (deterministic) regret of Algorithm 2 is upper bounded by $1/2$.*

*Proof sketch.* The deterministic case is immediate: by posting $1/2$, the regret in the first round is at most $1/2$, and the algorithm pays no regret in the subsequent rounds. For the stochastic case, due to Lemma 1, note that $\frac{1}{t}\sum_{i=1}^{t} \text{FGFT}(p, S_t, B_t) = \int_0^1 \frac{1}{t}\sum_{i=1}^{t} \mathbb{I}\{S_i \le p - u\}\mathbb{I}\{p + u \le B_i\}\,\mathrm{d}u =: \widehat{G}_t(p)$. Now, by the two-dimensional DKW inequality, it can be proved that the empirical estimates $\widehat{G}_t(p)$ are uniformly $\varepsilon$-close (in $p$) to $\mathbb{E}\big[\text{FGFT}(p, S_{t+1}, B_{t+1})\big]$ with probability $1 - O\big(e^{-\Omega(\varepsilon^2 t)}\big)$. These probability estimates, together with the fact that the algorithm selects $P_{t+1}$ by maximizing $p \mapsto \widehat{G}_t(p)$, translates to a $O\big(1/\sqrt{t+1}\big)$-control over the expectation $\max_{0 \le p \le 1} \mathbb{E}\big[\text{FGFT}(p, S_{t+1}, B_{t+1}) - \text{FGFT}(P_{t+1}, S_{t+1}, B_{t+1})\big]$. The conclusion follows by summing over time steps. $\qquad\square$

We now show that the guarantees provided by Algorithm 2 are worst-case optimal, even if sellers' and buyers' valuations are required to be independent of each other.

**Theorem 7.** *There exists a constant $c > 0$ such that the following holds. For every algorithm for the fair bilateral trade problem with full feedback and for every time horizon $T \in \mathbb{N}$, there exists an i.i.d. sequence $(S_t, B_t)_{t \in [T]}$ of pairs of sellers' and buyers' valuations such that, for each $t \in [T]$, $S_t$ is independent of $B_t$, and the algorithm suffers regret at least $c\sqrt{T}$ on that sequence.*

Since the proof of this result follows along the same lines as the proof of Theorem 3 (whose full details can be found in Appendix A), we present only a proof sketch.

*Proof sketch.* The hard instances are the same as those in the proof of Theorem 3 for the $\Omega(T^{2/3})$ lower bound in the stochastic i.i.d. case with independent sellers' and buyers' valuations. We refer to the proof sketch of Theorem 3 for the relevant notation and observations. We recall that in those instances to avoid suffering $\Omega\big(|\varepsilon|T\big)$ regret we need to distinguish the sign of $\varepsilon$. Now, given that full-feedback is available, the information we retrieve after each interaction is equivalent to observing a Bernoulli r.v. of parameter $\frac{1+\varepsilon}{2}$, regardless of the price we posted. Again, information-theoretic arguments imply that we need $\Omega\big(\frac{1}{\varepsilon^2}\big)$ rounds before being able to distinguish the sign of $\varepsilon$. During these rounds, the best we can do is to play (essentially) at random in the candidate set of optimal prices $\{1/2, 5/8\}$, suffering an expected instantaneous regret of $\Omega(|\varepsilon|)$. Overall, we suffer $\Omega\big(\min\big\{|\varepsilon| \cdot \frac{1}{\varepsilon^2}, |\varepsilon|T\big\}\big)$ cumulative regret, which leads to the claimed $\Omega(\sqrt{T})$ regret rate once we tune $|\varepsilon| = \Theta\Big(\sqrt{\frac{1}{T}}\Big)$. $\qquad\square$

**Remark 3.** *In the same spirit of Remark 1, Theorem 7 together with Yao's Minimax Theorem immediately imply that any* full-feedback *algorithm to solve the* adversarial *fair bilateral trade problem has to suffer regret of at least $\Omega(\sqrt{T})$. A nearly matching (up to logarithmic factors) upper bound for this problem can be deduced using the algorithm Hedge for $[0,1]$-Armed Experts (see [10, Appendix A]). We leave to future research the understanding of whether the adversarial case is logarithmically harder than the stochastic case or whether a different algorithm can achieve better regret rates.*

## 6 Limitations and conclusions

Our analysis is based on different assumptions on the generation of the sellers' and buyers' valuations (stochastic, stochastic and independent, deterministic, adversarial). Our results characterize (up to constant or log factors) the regret rates in each case. Hence, there are no specific *a priori* assumptions that we need to make to prove our results.

Our results can be extended in different directions. For example, by linking the unknown valuations of sellers and buyers to contextual information visible to the platform. In practice, platforms simultaneously deal with multiple sellers and buyers, in which case the platform may have to post a set of prices and consequently operate in a multidimensional decision space. Finally, our result for the stochastic setting with independence between sellers and buyers is only tight up to logarithmic factors, so an improved analysis of the upper or lower bound is needed.

## Acknowledgments and Disclosure of Funding

The work of FB was supported by the Project GAP (ANR-21-CE40-0007) of the French National Research Agency (ANR) and by the Chair UQPhysAI of the Toulouse ANITI AI Cluster. NCB and RC are partially supported by the MUR PRIN grant 2022EKNE5K (Learning in Markets and Society), the FAIR (Future Artificial Intelligence Research) project, funded by the NextGenerationEU program within the PNRR-PE-AI scheme, the EU Horizon CL4-2022-HUMAN-02 research and innovation action under grant agreement 101120237, project ELIAS (European Lighthouse of AI for Sustainability). TC gratefully acknowledges the support of the University of Ottawa through grant GR002837 (Start-Up Funds) and that of the Natural Sciences and Engineering Research Council of Canada (NSERC) through grants RGPIN-2023-03688 (Discovery Grants Program) and DGECR2023-00208 (Discovery Grants Program, DGECR - Discovery Launch Supplement).

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

## A  Proof of Theorem 3

Fix any $\varepsilon \in [0, 1/4]$ and let

$$\mu_{\pm\varepsilon} \coloneqq \frac{1 \pm \varepsilon}{2}\delta_0 + \frac{1 \mp \varepsilon}{2}\delta_{1/4}$$

where $\delta_a$ is the Dirac measure centered at a point $a \in \mathbb{R}$. Noting that, for all $p \in [0, 1]$,

$$p \le 1 - p \iff p \le \frac{1}{2}$$

$$p - \frac{1}{4} \le 1 - p \iff p \le \frac{5}{8}$$

and letting $S^{\pm}$ be a random variable with distribution $\mu_{\pm}$ and $B = 1$, we have

$$f^{\pm}(p) \coloneqq \mathbb{E}\Big[\min\big\{(p - S^{\pm})_+, (B - p)_+\big\}\Big] = \mathbb{E}\Big[\min\big\{(p - S^{\pm})_+, 1 - p\big\}\Big]$$

$$= \min\{p, 1 - p\}\frac{1 \pm \varepsilon}{2} + \min\Big\{\Big(p - \frac{1}{4}\Big)_+, 1 - p\Big\}\frac{1 \mp \varepsilon}{2}$$

$$= \begin{cases} \frac{1\pm\varepsilon}{2}p & p \in [0, 1/4) \\ \frac{1\pm\varepsilon}{2}p + \big(p - \frac{1}{4}\big)\frac{1\mp\varepsilon}{2} & p \in [1/4, 1/2) \\ (1 - p)\frac{1\pm\varepsilon}{2} + \big(p - \frac{1}{4}\big)\frac{1\mp\varepsilon}{2} & p \in [1/2, 5/8) \\ (1 - p)\frac{1\pm\varepsilon}{2} + (1 - p)\frac{1\mp\varepsilon}{2} & p \in [5/8, 1] \end{cases}$$

$$= \begin{cases} \frac{1\pm\varepsilon}{2}p & p \in [0, 1/4) \\ \frac{1\pm\varepsilon}{8} + \big(p - \frac{1}{4}\big) & p \in [1/4, 1/2) \\ \big(\frac{1\pm\varepsilon}{8} + \frac{1}{4}\big) \pm \varepsilon\big(\frac{1}{2} - p\big) & p \in [1/2, 5/8) \\ \big(\frac{1\pm\varepsilon}{8} + \frac{1}{4} \mp \frac{\varepsilon}{8}\big) + \big(\frac{5}{8} - p\big) & p \in [5/8, 1] \end{cases}$$

Note that:

$$\operatorname{argmax} f^+ = \frac{1}{2} \quad \text{and} \quad f^+\Big(\frac{1}{2}\Big) = \frac{1 + \varepsilon}{8} + \frac{1}{4} = \frac{3 + \varepsilon}{8}$$

$$\operatorname{argmax} f^- = \frac{5}{8} \quad \text{and} \quad f^-\Big(\frac{5}{8}\Big) = \frac{1 - \varepsilon}{8} + \frac{1}{4} + \frac{\varepsilon}{8} = \frac{3}{8}$$

Consider the three regions $I^{\mathrm{sub}} = [0, 1/4)$, $I^+ = [1/4, 9/16)$, $I^- = (9/16, 1]$. At a high level: $I^{\mathrm{sub}}$ is an informative, but $\Theta(1)$-suboptimal region; in contrast, regions $I^+$ and $I^-$ are both uninformative, but, when the underlying distribution of the sellers' valuations is $\mu_{+\varepsilon}$ (resp, $\mu_{-\varepsilon}$) $I^+$ (resp, $I^-$) contains the maximizer $p^{\star}_+ = \frac{1}{2}$ (resp, $p^{\star}_- = \frac{5}{8}$) while all points is $I^-$ (resp., $I^+$) are $\Theta(\varepsilon)$-suboptimal. More precisely:

- Since $B = 1$, the feedback $\mathbb{I}\{p \le B\}$ is immaterial.
- The feedback $\mathbb{I}\{S^{\pm}_t \le p\}$ observed by posting a price $p \in I^{\mathrm{sub}}$ at a time $t \in \mathbb{N}$ is a Bernoulli random variable with parameter $\frac{1\pm\varepsilon}{2}$. Moreover, for all $p \in I^{\mathrm{sub}}$

$$\max f^+ - f^+(p) = f^+\Big(\frac{1}{2}\Big) - f^+(p) = \frac{3 + \varepsilon}{8} - \frac{1 + \varepsilon}{2}p \ge \frac{1}{4}$$

$$\max f^- - f^-(p) = f^-\Big(\frac{5}{8}\Big) - f^-(p) = \frac{3}{8} - \frac{1 - \varepsilon}{2}p \ge \frac{1}{4}$$

  At a high level, posting a price $p \in I^{\mathrm{sub}}$ reveals information about the underlying distribution of the sellers' valuations but at the cost of an instantaneous regret of at least $1/4$.

- The feedback $\mathbb{I}\{S^{\pm}_t \le p\}$ observed by posting a price $p \in I^+$ at a time $t \in \mathbb{N}$ when the underlying sellers' distribution is $\mu_{\pm\varepsilon}$ is a constant random variable always equal to $1$. Moreover, $\operatorname{argmax} f^+ = \frac{1}{2} \in I^+$, but, for all $p \in I^+$,

$$\max f^- - f^-(p) = \begin{cases} \frac{3}{8} - \big(\frac{1-\varepsilon}{8} + \big(p - \frac{1}{4}\big)\big) \ge \frac{\varepsilon}{8} \ge \frac{\varepsilon}{16} & \text{if } p \in [1/4, 1/2) \\ \frac{3}{8} - \big(\big(\frac{1-\varepsilon}{8} + \frac{1}{4}\big) - \varepsilon\big(\frac{1}{2} - p\big)\big) \ge \frac{\varepsilon}{16} & \text{if } p \in [1/2, 9/16) \end{cases}$$

  At a high level, posting a price $p \in I^+$ reveals no information about the underlying distribution of the sellers' valuations, the region $I^+$ contains the optimal price when the underlying distribution of the sellers' valuations is $\mu_{+\varepsilon}$, but posting in $I^+$ has an instantaneous regret of at least $\varepsilon/16$ when the underlying distribution of the sellers' valuations is $\mu_{-\varepsilon}$.

- The feedback $\mathbb{I}\{S_t^{\pm} \leq p\}$ observed by posting a price $p \in I^-$ at a time $t \in \mathbb{N}$ when the underlying sellers' distribution is $\mu_{\pm\varepsilon}$ is a constant random variable always equal to 1. Moreover, $\operatorname{argmax} f^- = \frac{5}{8} \in I^-$, but, for all $p \in I^-$,

$$\max f^+ - f^+(p) = \begin{cases} \frac{1+\varepsilon}{8} + \frac{1}{4} - \left(\left(\frac{1+\varepsilon}{8} + \frac{1}{4}\right) + \varepsilon\left(\frac{1}{2} - p\right)\right) \geq \frac{\varepsilon}{16} & \text{if } p \in [9/16, 5/8) \\ \frac{1+\varepsilon}{8} + \frac{1}{4} - \left(\left(\frac{1+\varepsilon}{8} + \frac{1}{4} - \frac{\varepsilon}{8}\right) + \left(\frac{5}{8} - p\right)\right) \geq \frac{\varepsilon}{8} \geq \frac{\varepsilon}{16} & \text{if } p \in [5/8, 1] \end{cases}$$

  At a high level, posting a price $p \in I^-$ reveals no information about the underlying distribution of the sellers' valuations, the region $I^-$ contains the optimal price when the underlying distribution of the sellers' valuations is $\mu_{-\varepsilon}$, but posting in $I^-$ has an instantaneous regret of at least $\varepsilon/16$ when the underlying distribution of the sellers' valuations is $\mu_{+\varepsilon}$.

We will now show that these two "+" and "−" instances are no easier than two corresponding instances of a related partial monitoring game (for the reader unfamiliar with partial monitoring, see, e.g, [27, Chapter 27]). Consider the following partial monitoring game, where, again $\varepsilon \in [0, 1/4]$, the learner's action set is $\{1, 2, 3\}$, the environment's outcome set is $\{1, 2\}$, the reward ($\rho$) and feedback ($\varphi$) matrices are, respectively,

$$\rho \coloneqq \begin{bmatrix} 0 & 0 \\ 1/16 & 0 \\ 0 & 1/16 \end{bmatrix} \quad \text{and} \quad \varphi \coloneqq \begin{bmatrix} 1 & 0 \\ 0 & 0 \\ 0 & 0 \end{bmatrix}$$

and where the environment is constrained to draw outcomes i.i.d. from one of two distributions: the first of which has a probability $\frac{1}{2} + \frac{\varepsilon}{2}$ of drawing outcome 1, while the second has a smaller probability of $\frac{1}{2} - \frac{\varepsilon}{2}$ of drawing outcome 1. More precisely, let $(A_1^+, A_1^-, A_2^+, A_2^-, \dots)$ be an independent sequence of $\{1, 2\}$-valued random variables such that $(A_t^+)_{t \in \mathbb{N}}$ is an i.i.d. sequence satisfying $\mathbb{P}[A_t^+ = 1] = \frac{1}{2} + \frac{\varepsilon}{2}$ for all $t \in \mathbb{N}$, and $(A_t^-)_{t \in \mathbb{N}}$ is an i.i.d. sequence satisfying $\mathbb{P}[A_t^- = 1] = \frac{1}{2} - \frac{\varepsilon}{2}$ for all $t \in \mathbb{N}$. In the "+" (resp, a "−") scenario, at any time $t$, the environment plays outcome $A_t^+$ (resp., $A_t^-$). Therefore, in the "+" scenario, the learner's optimal action is action 2, and, for all $t \in \mathbb{N}$, the instantaneous expected regrets paid by the learner for playing action 1 (resp., 3) is $\mathbb{E}[\rho(2, A_t^+)] - \mathbb{E}[\rho(1, A_t^+)] = \frac{1}{32} + \frac{\varepsilon}{32} \leq \frac{1}{4}$ (resp., $\mathbb{E}[\rho(2, A_t^+)] - \mathbb{E}[\rho(3, A_t^+)] = \left(\frac{1}{32} + \frac{\varepsilon}{32}\right) - \left(\frac{1}{32} - \frac{\varepsilon}{32}\right) = \frac{\varepsilon}{16}$). Similarly, in the "−" scenario, the learner's optimal action is action 3, and, for all $t \in \mathbb{N}$, the instantaneous expected regrets paid by the learner for playing action 1 (resp., 2) is $\mathbb{E}[\rho(3, A_t^-)] - \mathbb{E}[\rho(1, A_t^-)] = \frac{1}{32} + \frac{\varepsilon}{32} \leq \frac{1}{4}$ (resp., $\mathbb{E}[\rho(3, A_t^-)] - \mathbb{E}[\rho(2, A_t^-)] = \left(\frac{1}{32} + \frac{\varepsilon}{32}\right) - \left(\frac{1}{32} - \frac{\varepsilon}{32}\right) = \frac{\varepsilon}{16}$). Finally, playing actions 2 or 3 reveals no feedback, while the feedback $\varphi(1, A_t^{\pm})$ that the learner observes after playing action 1 at time $t \in \mathbb{N}$ in scenario "$\pm$" is a Bernoulli random variable with parameter $\frac{1}{2} \pm \frac{\varepsilon}{2}$.

Note now that every algorithm for the fair bilateral trade problem can be turned into an algorithm for this partial monitoring game by simply mapping the price $P_t$ that the algorithm would post at any time $t \in \mathbb{N}$ into the action

$$J_t = \begin{cases} 1 & \text{if } P_t \in I^{\mathrm{sub}} \\ 2 & \text{if } P_t \in I^+ \\ 3 & \text{if } P_t \in I^- \end{cases}$$

then feeding back to the algorithm the pair $(\varphi(I_t, A_t^{\pm}), 1)$. By construction, this is the same feedback (more precisely, it is a Bernoulli random variable drawn i.i.d. from the same distribution) that the algorithm would observe by playing $P_t$ in the original fair bilateral trade problem, and the expected regret of the partial monitoring game is less than or equal to that of the original fair bilateral trade problem. Therefore, by lower bounding the regret of the partial monitoring game, we *a fortiori* lower bound the regret of the fair bilateral trade problem. To conclude the proof, we simply remark that for any algorithm for this partial monitoring game there exists a choice of $\varepsilon \in [0, 1/4]$ such that the regret of the algorithm is at least $T^{2/3}/112$ (see, e.g., [16, Theorem 5.1], up to turning losses into rewards and rescaling them by multiplying by $1/16$).

# B  Proof of Theorem 4

Given that we are in a deterministic setting, without loss of generality we (can) restrict our analysis to deterministic algorithms. Let $b = 1$. We begin by proving the following property by induction on $k = 0, 1, 2, \dots$.

**Property.** There is a segment $E_k$ of length $\frac{1}{4}2^{-k}$ satisfying $E_k \subset \left[0, \frac{1}{4}\right]$ such that, for each $s, s' \in E_k$, the algorithm receives the same feedback from the environments defined by $(s, b)$ and $(s', b)$—and thus selects the same query prices—as long as it has allocated at most $k$ prices in the segment $\left[0, \frac{1}{4}\right]$.

For $k = 0$ the property is true by setting $E_0 := \left[0, \frac{1}{4}\right]$, since as long as the algorithm posts no prices in $\left[0, \frac{1}{4}\right]$, it receives for each price $P_t \in (\frac{1}{4}, 1]$ the feedback $\mathbb{I}\{s \le P_t\} = \mathbb{I}\left\{\frac{1}{4} \le P_t\right\} = 1$ and $\mathbb{I}\{P_t \le b\} = \mathbb{I}\{P_t \le 1\} = 1$.

Assume that the property is true for some $k \in \{0, 1, 2, \dots\}$. We set $[c, d] := E_k$ for the segment of the property. For any $s \in E_k$, the algorithm has a behavior that does not depend on $s$ until it posts a price $P_t \in \left[0, \frac{1}{4}\right]$ for the $(k + 1)$-th time, since before that it has received feedback that does not depend on $s$. If the algorithm never posts a price $P_t \in \left[0, \frac{1}{4}\right]$ for the $(k + 1)$-th time, then the property remains true (e.g., by setting $E_{k+1} := \left[\frac{c+d}{2}, d\right]$). If the algorithm posts this price $P_t$ for the $(k + 1)$-th time, we set $E_{k+1} := \left[\frac{c+d}{2}, d\right]$ if $P_t < \frac{c+d}{2}$ and we set $E_{k+1} := \left[c, \frac{c+d}{2}\right]$ if $P_t \ge \frac{c+d}{2}$. Note that this price $P_t$ is the same for any $s \in E_{k+1}$, since it depends only on the feedback from times 1 to $t - 1$. At this time $t$, for any $s \in E_{k+1}$, the feedback is the same. Indeed, regardless of which $s \in E_{k+1}$ we might choose, this feedback is $\left(\mathbb{I}\{s \le P_t\}, \mathbb{I}\{P_t \le b\}\right) = (0, 1)$ if $P_t < \frac{c+d}{2}$, while it is $\left(\mathbb{I}\{s \le P_t\}, \mathbb{I}\{P_t \le b\}\right) = (1, 1)$ if $P_t \ge \frac{c+d}{2}$. The feedback then remains the same for all the next times until a time $t' > t$ when the algorithm posts a price $P_{t'} \in \left[0, \frac{1}{4}\right]$ for the $(k + 2)$-th time. Hence the property is true at step $k + 1$. Note that our construction shows that we can even consider nested segments $E_0 \supseteq E_1 \supseteq E_2 \supseteq \cdots$.

There is a value $k^\star$ of $k \in \{0, \dots, T\}$ such that the property is true, and for any $s \in E_{k^\star}$, the algorithm posts exactly $k^\star$ times a price in $\left[0, \frac{1}{4}\right]$ from times 1 to $T$. This is because if we write $t_0 = 0$ and for $k \ge 1$, $t_k$ for the time at which the algorithm posts for the $k$-th time a price in $\left[0, \frac{1}{4}\right]$ (for any $s \in E_k$), then we have $t_0 \le t_1 \le \cdots$. Hence, we let $k^\star$ be the largest $k$ such that $t_k \le T$.

For any $s \in E_{k^\star}$ the instantaneous regret for the time steps where $P_t \in \left[0, \frac{1}{4}\right]$ is lower bounded as

$$\frac{b - s}{2} - (P_t - s)_+ \ge \frac{3}{8} - \frac{1}{4} = \frac{1}{8} \ .$$

Then let $\mathcal{T}$ be the set of time steps in $\{1, \dots, T\}$ when $P_t \in (\frac{1}{4}, 1]$. Note that $(P_t)_{t \in \mathcal{T}}$ does not depend on which $s \in E_{k^\star}$ we might have chosen. The FGFT function reaches its maximum value $\frac{b-s}{2}$ at $\frac{b+s}{2} \in \left[\frac{1}{2}, \frac{5}{8}\right]$, and decreases with slope one in the segments $\left[\frac{1}{4}, \frac{b+s}{2}\right]$ and $\left[\frac{b+s}{2}, 1\right]$. Hence, the FGFT function at any $p \in (\frac{1}{4}, 1]$ is equal to $\frac{b-s}{2} - \left|p - \frac{b+s}{2}\right|$.

Hence the cumulative regret for the steps in $\mathcal{T}$ is lower bounded by

$$\sum_{t \in \mathcal{T}} \left|P_t - \frac{b + s}{2}\right| \ge \left|\sum_{t \in \mathcal{T}} \left(P_t - \frac{b + s}{2}\right)\right| = |\mathcal{T}| \left|\left(\frac{1}{|\mathcal{T}|} \sum_{t \in \mathcal{T}} P_t - \frac{b}{2}\right) - \frac{s}{2}\right| \ .$$

Since the prices $P_1, \dots, P_T$ chosen by the algorithm are the same for any $s \in E_{k^\star}$, then $\frac{1}{|\mathcal{T}|} \sum_{t \in \mathcal{T}} P_t - \frac{b}{2}$ does not depend on $s$ and thus we can move $s$ in the segment of length $\frac{1}{4}2^{-k^\star}$, and find a value of $s$ for which $\left|\left(\frac{1}{|\mathcal{T}|} \sum_{t \in \mathcal{T}} P_t - \frac{b}{2}\right) - \frac{s}{2}\right| \ge \frac{1}{16}2^{-k^\star}$. Hence, for this $s$ the cumulative regret of the steps in $\mathcal{T}$ is lower bounded by

$$|\mathcal{T}| \cdot \frac{1}{16}2^{-k^\star} \ .$$

In the end, the total cumulative regret is lower bounded by

$$\frac{k^\star}{8} + \frac{1}{16}(T - k^\star)2^{-k^\star} \ .$$

If $k^\star \ge \frac{T}{2}$, we can lower bound the total cumulative regret by $\frac{T}{16}$. Else, we can lower bound the total cumulative regret by

$$\frac{k^\star}{8} + \frac{1}{32}T2^{-k^\star} \ .$$

If $2^{k^\star} \le \frac{T}{\log_2(T)}$ then the total cumulative regret is lower bounded by $\frac{\log_2(T)}{32}$. Else, we have

$$2^{k^\star} \ge \frac{T}{\log_2(T)} \ge \sqrt{T} \ ,$$

where in the last inequality we used $T \geq 17$, and hence $k^* \geq \log_2(\sqrt{T}) = \frac{\log_2(T)}{2}$. In the end, the lower bound is

$$\min\left\{\frac{T}{16}, \frac{\log_2(T)}{32}\right\} = \frac{\log_2(T)}{32}.$$

## C Pseudocode of Double Binary Search Pricing and proof of Theorem 5

The pseudocode of Double Binary Search Pricing is given in Algorithm 3. In the algorithm and in the proof, we use the following notation: for a segment of the form $E := [c, d]$ with $c < d$, we define $\text{left}(E) := c$, $\text{mid}(E) := \frac{c+d}{2}$ and $\text{right}(E) := d$.

---

**Algorithm 3:** Double Binary Search Pricing - Deterministic setting

---

**Input**: horizon $T \in \mathbb{N}$;
**Initialization**: Set $E_S := E_B := [0, 1]$. If $2\lceil\log_2(T)\rceil + 1 \leq T$ set $N := \lceil\log_2(T)\rceil$, else set $N := 0$;
**for** *time $t = 1, 2, \ldots, N$* **do**
    Post price $P_t := \text{mid}(E_S)$;
    **if** $s \leq P_t$ **then**
        Set $E_S := [\text{left}(E_S), \text{mid}(E_S)]$;
    **else**
        Set $E_S := [\text{mid}(E_S), \text{right}(E_S)]$;
**for** *time $t = N + 1, N + 2, \ldots, 2N$* **do**
    Post price $P_t := \text{mid}(E_B)$;
    **if** $P_t \leq b$ **then**
        Set $E_B := [\text{mid}(E_B), \text{right}(E_B)]$;
    **else**
        Set $E_B := [\text{left}(E_B), \text{mid}(E_B)]$;
**for** *time $t = 2N + 1, \ldots, T$* **do**
    Post price $P_t := \frac{\text{mid}(E_S)+\text{mid}(E_B)}{2}$;

---

We now present the full proof of Theorem 5.

*Proof.* If $T < 2\lceil\log_2(T)\rceil + 1$, the result is true because FGFT takes values in $[0, 1]$ and so the instantaneous regret at any time is bounded by 1. Consider then the case $2\lceil\log_2(T)\rceil + 1 \leq T$.

The cumulative regret from time steps 1 to $2N$ is bounded by $2N = 2\lceil\log_2(T)\rceil$ because FGFT $\in [0, 1]$. It is easy to see that the property that $s \in E_S$ is maintained from time steps 1 to $N$. At time $N$ (after which $E_S$ is left unchanged), the length of $E_S$ is smaller than or equal to $2^{-N} \leq \frac{1}{T}$. Similar properties hold for $b$ and $E_B$. Hence, at time steps $2N + 1, \ldots, T$, we have that

$$\left|\frac{\text{mid}(E_S) + \text{mid}(E_B)}{2} - \frac{s+b}{2}\right| \leq \frac{1}{2}\big(|s - \text{mid}(E_S)| + |b - \text{mid}(E_B)|\big) \leq \frac{1}{T}. \tag{6}$$

Therefore, the individual regret at each one of these time steps is bounded by

$$\frac{b-s}{2} - \min\left\{\left(\frac{\text{mid}(E_S) + \text{mid}(E_B)}{2} - s\right)_+, \left(b - \frac{\text{mid}(E_S) + \text{mid}(E_B)}{2}\right)_+\right\}. \tag{7}$$

If $b - s \leq \frac{2}{T}$ this quantity is smaller than or equal to $\frac{1}{T}$. If $b - s > \frac{2}{T}$, then from (6), we have

$$\frac{\text{mid}(E_S) + \text{mid}(E_B)}{2} - s \geq \frac{b-s}{2} - \left|\frac{\text{mid}(E_S) + \text{mid}(E_B)}{2} - \frac{s+b}{2}\right| \geq \frac{b-s}{2} - \frac{1}{T}.$$

Similarly

$$b - \frac{\text{mid}(E_S) + \text{mid}(E_B)}{2} \geq \frac{b-s}{2} - \left|\frac{\text{mid}(E_S) + \text{mid}(E_B)}{2} - \frac{s+b}{2}\right| \geq \frac{b-s}{2} - \frac{1}{T}.$$

Hence, the quantity in 7 is smaller than or equal to $\frac{1}{T}$, and consequently, the cumulative regret from steps $2N + 1$ to $T$ is smaller than or equal to $\frac{T-2N}{T} \leq 1$. Putting everything together, the cumulative regret from steps 1 to $T$ is smaller than or equal to $1 + 2\lceil\log_2(T)\rceil$. $\qquad\square$

## D    Proof of Theorem 6

For notational convenience, let $(S, B)$ be another pair of seller/buyer valuations, independent of the whole sequence $(S_t, B_t)_{t\in\mathbb{N}}$, and sharing the same distribution of any element in this i.i.d. sequence. The proof shares ideas with the proof of [9, Theorem 1]. For each $p \in [0, 1]$, notice that, as a consequence of Equation (1) in Lemma 1, we have

$$\mathbb{E}\big[\text{FGFT}(p, S, B)\big] = \int_0^1 \mathbb{P}\big[\{S \le p - u\} \cap \{p + u \le B\}\big]\,\mathrm{d}u =: G(p)\,,$$

and, for each $t \in \mathbb{N}$, also

$$\frac{1}{t}\sum_{i=1}^t \text{FGFT}(p, S_t, B_t) = \int_0^1 \frac{1}{t}\sum_{i=1}^t \mathbb{I}\{S_i \le p - u\}\mathbb{I}\{p + u \le B_i\}\,\mathrm{d}u =: \widehat{G}_t(p)$$

We have, for any $p \in [0, 1]$ and any $t \in \mathbb{N}$,

$$\big|G(p) - \widehat{G}_t(p)\big| = \left|\int_0^1 \mathbb{P}\big[\{S \le p - u\} \cap \{-B \le -p - u\}\big]\,\mathrm{d}u - \int_0^1 \frac{1}{t}\sum_{i=1}^t \mathbb{I}\{S_i \le p - u\}\mathbb{I}\{-B_i \le -p - u\}\,\mathrm{d}u\right|$$

$$\le \sup_{x,y\in\mathbb{R}}\left|\mathbb{P}\big[\{S \le x\} \cap \{-B \le y\}\big] - \frac{1}{t}\sum_{i=1}^t \mathbb{I}\{S_i \le x, -B_i \le y\}\right|.$$

Hence, we can apply the two-dimensional DKW inequality, see [9, Theorem 15, Appendix J], from which it follows that there are positive constants $m_0 \le 1200$, $c_1 \le 13448$ and $c_2 \ge 1/576$ such that for all $\varepsilon > 0$, if $t \ge m_0/\varepsilon^2$,

$$\mathbb{P}\big[\big|G(p) - \widehat{G}_t(p)\big| \ge \varepsilon\big] \le \mathbb{P}\left[\sup_{x,y\in\mathbb{R}}\left|\mathbb{P}\big[\{S \le x\} \cap \{-B \le y\}\big] - \frac{1}{t}\sum_{i=1}^t \mathbb{I}\{S_i \le x, -B_i \le y\}\right| \ge \varepsilon\right] \le c_1 e^{-c_2 t\varepsilon^2}.$$

Hence, for each $t \in \mathbb{N}$, by Fubini's theorem, we have that

$$\mathbb{E}\left[\sup_{p\in[0,1]}\big|G(p) - \widehat{G}_t(p)\big|\right] = \int_0^\infty \mathbb{P}\left[\sup_{p\in[0,1]}\big|G(p) - \widehat{G}_t(p)\big| \ge \varepsilon\right]\,\mathrm{d}\varepsilon \le \sqrt{\frac{m_0}{t}} + \int_{\sqrt{\frac{m_0}{t}}}^\infty c_1 \exp(-c_2 t\varepsilon^2)\,\mathrm{d}\varepsilon$$

$$\le \sqrt{\frac{m_0}{t}} + \frac{c_1}{2}\int_{-\infty}^\infty \exp(-c_2 t\varepsilon^2)\,\mathrm{d}\varepsilon = \sqrt{\frac{m_0}{t}} + \frac{c_1}{2}\frac{\sqrt{2\pi}}{\sqrt{2c_2 t}} = \left(\sqrt{m_0} + \frac{c_1\sqrt{\pi}}{2\sqrt{c_2}}\right)\frac{1}{\sqrt{t}}.$$

Then, for any $t \in \mathbb{N}$, we have, using the law of total expectation conditioned on $(S_i, B_i)_{i=1,\dots,t}$,

$$\mathbb{E}\big[\text{FGFT}(p^\star, S_{t+1}, B_{t+1})\big] - \mathbb{E}\big[\text{FGFT}(P_{t+1}, S_{t+1}, B_{t+1})\big] = G(p^\star) - \mathbb{E}\big[G(P_{t+1})\big]$$

$$= G(p^\star) - \mathbb{E}\big[\widehat{G}_t(p^\star)\big] + \mathbb{E}\big[\widehat{G}_t(p^\star) - \widehat{G}_t(P_{t+1})\big] + \mathbb{E}\big[\widehat{G}_t(P_{t+1}) - G(P_{t+1})\big]$$

$$\le G(p^\star) - \mathbb{E}\big[\widehat{G}_t(p^\star)\big] + \mathbb{E}\big[\widehat{G}_t(P_{t+1}) - G(P_{t+1})\big] \le 2\mathbb{E}\left[\sup_{p\in[0,1]}\big|\widehat{G}(p) - G(p)\big|\right] \le 2\left(\sqrt{m_0} + \frac{c_1\sqrt{\pi}}{2\sqrt{c_2}}\right)\frac{1}{\sqrt{t}}\,.$$

Hence, for any $T \ge 2$,

$$R_T \le 1 + 2\left(\sqrt{m_0} + \frac{c_1\sqrt{\pi}}{2\sqrt{c_2}}\right)\sum_{t=2}^T \frac{1}{\sqrt{t-1}} \le 1 + 4\left(\sqrt{m_0} + \frac{c_1\sqrt{\pi}}{2\sqrt{c_2}}\right)\sqrt{T-1}\,.$$

