# OpenReview forum: "Fair Online Bilateral Trade"
_NeurIPS.cc/2024/Conference — NeurIPS 2024 poster_

### Official Review · Reviewer_rYk6 · 2024-06-20

**Soundness:** 4
**Presentation:** 4
**Contribution:** 4
**Rating:** 8
**Confidence:** 4

**Summary:**

This paper addressed the problem of fair bilateral trade: at each round $t\leq T$ a buyer and a seller, with respective valuations $B_t$ and $S_t$, want to trade a good. The agent acts as a facilitator for the trade, by posting a common price $p_t$. The trade happens if $p_t \leq B_t$ and  $p_t \geq S_t$. Importantly, the agent only observes the two-bit feedback $(\1{p_t \leq B_t\}, p_t \geq S_t)$.

In classical bilateral trade, the objective is to maximize the cumulative expected Gain From Trade (GFT), where the GFT at round $t$ is $B_t - S_t$ if the trade succeeds, 0 otherwise. Here, the authors propose to consider an other performance measure called the Fair Gain From Trade, defined as $\min(B_t-p_t,p_t - S_t)$ if the trade happens, 0 otherwise. This performance measures encourages trades that share fairly the utility between buyer and seller.

The authors first consider the two-bit feedback model with i.i.d. valuations of for the buyer and the seller. They show that, as is the case when maximizing the GFT, the FGFT can be linear in $T$ if the seller's and the buyer's valuations are not independent. Interestingly, when these valuations are independent, regrets of order $\tilde{O}(T^{2/3})$ (in terms of FGFT) can be achieved under milder assumption than when considering GFT: namely, Lipschitz continuity of the c.d.f. of the valuations is no longer required. They provide a matching lower bound, proving that this rate is tight up to logarithmic factors. They also consider deterministic valuations, showing that in this case, the regret scales as $\log(T)$ (they provide an upper and lower bound on the regret). Again, this departs from the regrets observed when maximizing GFT. Finally, they consider the full information model, when the agent observes both $B_t$ and $S_t$. They show that in this case, the regret scales as $\sqrt(T)$ in the stochastic case, which is optimal, and is constant in the deterministic case.

**Strengths:**

This paper studies an interesting an well-motivated problem. The results provided highlight a very different behavior than in classical bilateral trade problems, which I find very interesting. The treatment of the subject is thorough: the authors explore various models, including two-bit and full feedback, as well as i.i.d. and deterministic valuations, and both independent and dependent valuations. For each case, they provide matching upper and lower bounds on the regret. They also comment on the adversarial case.

The paper is also very clear and well-written. Each theorem is accompanied by easily understandable proof sketches, supplemented by rigorous proofs in the Appendix.

**Weaknesses:**

There are no major weaknesses.

As a minor suggestion, given the paper's density, it might be helpful for readers if the rates in the different settings were summarized in a table, alongside the corresponding rates for maximizing the gain from trade.

**Questions:**

The problem of bilateral trade is closely related to that of dynamic pricing. Do you know any work studying fairness in that setting?

Although I understand that this is probably beyond the scope of this already dense paper, do you think your results could be extended to the one-bit feedback setting?

**Limitations:**

The authors addressed adequately the limitations of their work.

---

> ### Author Rebuttal · Authors · 2024-08-06
>
> - Adding a summary table
>
> Great idea! We are happy to do it.
>
> - Fair dynamic pricing
>
> The few existing works on fair dynamic pricing study notions of fairness that are orthogonal to what ours would be when translated to dynamic pricing.
> For example, see Xu et al., Doubly Fair Dynamic Pricing, and Maestre et al., Reinforcement Learning for Fair Dynamic Pricing.
> Note that in bilateral trade, since the platform is the learner, there is a symmetry between the buyer's and seller's utility, and thus, it is sensible that our fair gain from trade function strives to make their utilities as similar as possible.
> In contrast, in dynamic pricing, the seller is the learner, and hence there is an inherent asymmetry in how the learner will choose their objective. The seller might still have an incentive to reduce their margin to improve that of the buyer because this way, they could provide enough incentives for buyers to join and continue using the platform.
> For this reason, we think that a suitable alternative reward function in dynamic pricing should remain somewhat asymmetric and give more weight to the seller's margin.
> An example can be the minimum between the seller's profit and a factor of $w$ times the buyer's profit.
> A similar idea was also discussed in the answer to reviewer 89Fg, in the context of bilateral trade.
>
> - One-bit feedback
>
> We conjecture that the techniques in [8] can be tweaked to show that learning is impossible with 1-bit feedback, for reasons analogous to that in [8]: lack of observability. With only 1 bit of feedback, we conjecture it is possible to build two different instances that are undistinguishable from one another such that the optimal price in one instance is highly suboptimal in the other.

---

> > ### Comment · Reviewer_rYk6 · 2024-08-07
> > **Rebuttal acknowledgment**
> >
> > The authors have adequately answered my questions.
> >
> > Having read again the grading guidelines for Neurips, I believe that this paper will belong to the 50% of the top accepted papers, and is a "clear accept". I therefore raise my score to 8.

---

### Official Review · Reviewer_89Fg · 2024-06-28

**Soundness:** 4
**Presentation:** 4
**Contribution:** 4
**Rating:** 7
**Confidence:** 4

**Summary:**

This paper studies the fair online bilateral trade problem, where a platform posts prices for one item at a time. At each time point, a (buyer, seller) pair arrives, each with private valuations. A trade occurs if the posted price is between the buyer and seller valuations. In this paper, the goal is to maximize the minimum utility of the buyer and the seller; this differs from prior work, where the objective was to maximize the total utility. The paper considers multiple valuation regimes, including deterministic, stochastic, stochastic and independent, and full feedback, and bounds the regret in each regime.

The main results of the paper are as follows. First, the paper provides a linear lower bound in the stochastic case where buyer and seller valuations are not necessarily independent. The paper complements this result with an upper bound of $\tilde{O}(T^{⅔})$ in the stochastic case where buyer and seller valuations are independent. This result relies on the Convolution Lemma, which is the key technical insight from the paper. The paper further shows that in the deterministic setting (where buyer and seller valuations are fixed over all rounds), there is a tight regret bound of $\Theta(\ln(T))$. Finally, when valuations are known, the paper provides an algorithm which achieves $O(\sqrt{T})$ regret and shows that this bound is tight.

**Strengths:**

The problem statement is interesting and the difference from prior work is clearly stated. The model is clean and easy to understand, and I appreciated the formal problem definition in lines 114 - 117. The paper provides comprehensive results (often both lower and upper bounds) for multiple reasonable assumptions, and gives intuition for where the difficulty comes from in each regime. The work seems relevant for NeurIPS.

I found the paper to be very easy to follow. The assumptions are clearly stated in each section, and the presentation is generally well done. In particular, I liked how the theorem statements are rigorous and clear and that proof sketches are given when there is a lack of space for full proofs. The proofs are nicely written and correct as far as I checked. In particular, Lemma 1 (the Convolution Lemma) was clean and clever, and is made good use of throughout the rest of the paper (i.e. in the proofs of Theorems 2 and 6). I felt that the lower bound examples in section 2 and in the proof of Theorem 1 were also excellent at building intuition for the remainder of the paper, and interesting examples in and of themselves.

**Weaknesses:**

The structure of section 1 made it a bit harder to follow than the rest of the paper. Personally, I would have found it helpful for section 1 to be split into multiple parts (such as intro/related work/our contributions) in order to be able to more easily pinpoint the main contributions.

While the paper presents a very nice theoretical model, it does not discuss many applications. I would have found it helpful if the paper had further discussed applications of fair online bilateral trade – for example, by discussing when such an objective function would make sense.

**Questions:**

How well and/or easily would the main results generalize to a weighted fair gain from trade objective? For example, would similar arguments work if the buyer’s utility was weighted twice as much as the seller’s utility?

What types of applications come to mind for the fair online bilateral trade problem?

**Limitations:**

The authors have adequately addressed the limitations.

---

> ### Author Rebuttal · Authors · 2024-08-06
>
> - Weighted fair gain from trade objective
>
> A possible weighted generalization of the fair gain from trade could be
>
> $WFGFT(p,s,b) = \min( w \cdot  (p-s)^+$, $(b - p)^+ )$
>
> for a fixed constant $w \ge 1$ and where we recall that $p$ is the posted price, $s$ is the seller valuation and $b$ is the buyer valuation.
> This would address the reviewer's question in the sense that here the buyer's utility (profit) is valued $w$ times that of the seller.
> For fixed $s <b$, the optimal price is  $(w s + b) / (w+1)$, and this price gets "pulled" toward the seller side as $w$ increases, thus increasing the buyer's profit (since the buyer's profit is valued more).
>
> Our regret upper bounds can be extended to this weighted fair gain from trade. This is straightforward in the deterministic setting, since we just need to locate the optimal price $(w s + b) / (w+1)$. We either see it immediately (in the full-feedback model) or, otherwise, we use a binary search.
>
> In the stochastic case, the following extension of the convolution lemma can be shown to hold (recall that it is under independence of buyer and seller):
>
> $\mathbb E [  WFGFT(p,S,B) ] = \int_0^1 \mathbb{P} [ S \le p - \frac{u}{w} ] \mathbb{P} [ u+p \le B ] du $
>
> Hence the proofs (Theorems 2 and 6) can be extended with minor adjustments, yielding the same rates.
>
> Analogous considerations hold for lower bounds, which can be easily adapted to obtain the same rates we obtained for $FGFT$.
>
> - Applications of fair online bilateral trade
>
> A natural application is online ride-sharing services like Uber and Lyft where fairness problems have been previously studied, although in different settings with metrics different from ours (see, e.g., Sühr et al. "Two-sided fairness for repeated matchings in two-sided markets: A case study of a ride-hailing platform", 2019).

---

> > ### Comment · Reviewer_89Fg · 2024-08-08
> >
> > I would like to thank the authors for their response, and have no further questions or comments at this time.

---

### Official Review · Reviewer_6Ak7 · 2024-07-10

**Soundness:** 3
**Presentation:** 3
**Contribution:** 3
**Rating:** 6
**Confidence:** 4

**Summary:**

The paper considers a fair version of online bilateral trade problem to minimize fair GFT regret w.r.t. optimal fixed price in hindsight.
The paper is comprehensive in studying both upper/lower bounds in various settings.

**Strengths:**

- The problem setup is pretty interesting, with a good motivation to the practical scenario.
- The paper is pretty comprehensive, and the paper well-written and easy to follow.
- It is interesting that ETC style of algorithm is order-optimal.

**Weaknesses:**

Technicality
- It is good to know that the ETC style of algorithm 1 works well, but I must admit that it is not particularly interesting.
- The algorithm for deterministic case is also tight, but the algorithm itself is a straightforward application of binary search, and the regret lower bound is not difficult to come up with, so I would exclude this section for my evaluation.


Minor comments
- Is it necessary to have bounded support of [0,1] for the valuations? If not (or it's wlog), it would be worth mentioning it.
- Why would one need to invoke minimax theorem to argue Remark 1? It seems to trivially follow as the adversary there is simply stronger, correct me if I'm wrong.
- It might be good to decompose the introduction to explicitly distinguish the true intro / the paper's results / relate work. As of now it's difficult to follow the intro as they are highly mixed up.
- L#80 does not seem necessary, or a footnote might suffice.
- The problem of online bilateral trade itself seems highly relevant to Kleinberg and Leighton FOCS'03 (Bounds on Regret for On-line Posted-Price Auctions) and its follow-up works, but it is never mentioned.
- Since the algorithm is rather simple but is proven to be order-optimal, I think it would be good to add experiments to see whether it works well compared to other candidates from standard bandit literature.

**Questions:**

- Has the one-shot version of this problem ever been studied? I.e., whether FGFT can be constant apx to the ex-post FGFT or any structural connection between FGFT, first-best GFT, etc.
- It seems the regret lower bound largely depends on a highly concentrated distribution. Do you think a sort of smoothness assumption like regularity or monotone hazard rates can be a remedy for better regret bounds

**Limitations:**

Mentioned above.

---

> ### Author Rebuttal · Authors · 2024-08-06
>
> - Weakness 1
>
> We are not sure we understand what the reviewer meant here.
> We are happy to provide clarifications if the reviewer needs some.
>
> - Valuations not in $[0,1]$
>
> Extending beyond $[0,1]$ works as in the bandit literature: If $[0,1]$ is replaced by $[0,m]$ (for some $m$), the same results apply up to a multiplicative $m$; If the interval is unbounded, no meaningful results can be obtained, unless additional assumptions on the distributions are made (e.g., $\sigma$-sub-Gaussianity, with known $\sigma$).
>
> - Why minimax?
>
> Yao's minimax theorem immediately implies the result, but the reviewer is correct in stating that the result would have also followed by observing that a lower bound on the expected regret of the stochastic case implies a lower bound in the adversarial case because in-expectation lower bounds can be turned into worst-case lower bounds.
> We will mention this in the revised version.
>
> - Kleinberg and Leighton (FOCS'03)
>
> We agree that dynamic pricing is a sufficiently related setting to be mentioned in the related work section.
> The revised version will contain a paragraph about it.
>
> - Experiments comparing against bandits
>
> The comparison with bandit algorithms would not be fair given that bandit algorithms cannot run in the two-bit feedback setting (the two bits of feedback are not sufficient to reconstruct the realized reward of posted prices, see Lines 39-41).
> Moreover, from a practical point of view, the bandit version of our problem is not well-motivated, as it is hard to imagine that a learner would be able to observe the reward of a price without access to the valuations (which would lead back to the full-info case).
> If the reviewer agrees with us, we would prefer not to implement this change.
>
> - One-shot version
>
> To the best of our knowledge, we are the first to introduce this fair variant of the bilateral trade problem, which arose as a natural continuation of the recent stream of works on online learning in bilateral trade.
> We agree with the reviewer that studying the one-shot version would also be an interesting extension of our work.
>
> - Does regularity help?
>
> It does not, in fact, our current lower bound constructions could be modified by "smoothing" the Dirac masses into uniform distributions over squares, obtaining the same results.
> To give some intuition as to why this is the case, in bilateral trade problems, smoothness helps by making the objective (i.e., the expected reward) Lipschitz, which in turn reduces the continuum-arm problem to a finite-arm problem via discretization.
> In the *fair* bilateral trade problem, however, the objective is already $1$-Lipschitz with no further assumptions.
> What hinders learnability in our problem is a lack of observability, which is recovered by assuming independence of buyers and sellers.
> Although we don't have an explicit lower bound construction at hand for the monotone hazard rate, we believe that this assumption would also be insufficient to obtain learnability.

---

> > ### Comment · Reviewer_6Ak7 · 2024-08-09
> >
> > Thanks the authors for the detailed response. I have no further question.
> > Although the techniques are not particularly interesting, I think the paper is worth being accepted as it introduces a novel problem that might interest the community in several directions.

---

### Official Review · Reviewer_mLA5 · 2024-07-12

**Soundness:** 4
**Presentation:** 3
**Contribution:** 2
**Rating:** 6
**Confidence:** 3

**Summary:**

The paper focuses on the online bilateral trade problem, in which at each round a buyer and a seller with private valuations for an item arrive, and the platform has to post prices for the item being traded. In this paper the objective of the platform is that of maximizing the cumulative “fair gain from trade”, that is the minimum between the seller’s and buyer’s utilities.

In the two-bit feedback setting, the paper proves a tight $O(\log T)$ regret bound in the deterministic setting, and a $O(T^{2/3)$ regret in the stochastic setting when seller’s and buyer’s valuations are independent of each other. In the full-feedback setting, the paper provides tight regret bounds.

**Strengths:**

The paper is well-structured and explores a research direction which could potentially be very interesting.

**Weaknesses:**

Section 2 is clear but does not sufficiently explain the additional challenges that need to be addressed compared to the standard online bilateral trade framework by Cesa-Bianchi et al. (EC 21).

The main area for improvement in this paper is the generalization of the results beyond the specific objective of Fair GFT. What is the general structure required to achieve results similar to those presented? Could similar results be directly obtained for other objectives? The justification for focusing specifically on maximizing Fair GFT is not particularly convincing at the moment. Developing a broader framework to address these problems would significantly strengthen the results and make them more widely applicable and interesting.

**Questions:**

See weaknesses

**Limitations:**

Limitations are discussed.

---

> ### Author Rebuttal · Authors · 2024-08-06
>
> - Additional challenges here compared to Cesa-Bianchi et al., EC '21
>
> A first high-level observation is that the pairs "assumption"/"regret rate" differ between our setting (Fair Bilateral Trade) and that of Cesa-Bianchi et al. [10] ("regular" Bilateral Trade), suggesting that different ideas will be needed to obtain optimal results for each pair.
> For example, while smoothness (bounded densities) was crucial in obtaining sublinear rates in [10], we had to understand that, in our work, it plays no role, and consequently discover new algorithmic and proof ideas.
> At a very high level, we share with [10] the same issue of having poor feedback, which is not even sufficient to reconstruct the reward at the price we post.
> In our case, however, the different form of the objective requires new procedures to recover usable information on the reward function, which is made possible by our new Convolution Lemma.
> Another difference is that, in [10], to maximize the realized gain from trade, it is sufficient to post a price $p\in[S,B]$, while, in our case, we have to address the more delicate task of locating the midpoint $p = (S+B)/2$.
> This is the reason why (even optimal) algorithms for bilateral trade can suffer linear regret in our setting (as we show in Section 2).
> Another technical subtlety that we had to tackle is that a direct application of the convolution lemma in the 2-bit feedback setting would yield a suboptimal upper bound of $T^{3/4}$.
> Instead, by carefully defining a data-gathering procedure such that each observation contributes to estimating the convolution of the cdfs at *all* points, we are able to obtain the optimal $T^{2/3}$ rate.
> We will better highlight these various challenges in the paper.
>
> - Justification for focusing specifically on maximizing Fair GFT
>
> We fully agree that developing a broader framework is an interesting goal to pursue. We suspect that obtaining estimators for a reasonably large class of objective functions in bilateral trade could be a challenging problem if one aims at capturing at least both the GFT and FGFT objectives.
> The main reason for this difficulty is that different objectives would likely require significantly different techniques due to the problem-specific quantities that the learner has to estimate.
> For example, the fair and regular gain from trade problems rely on different algorithms, estimation lemmas, proofs, and assumptions, as discussed above.
>
> Our specific focus on the Fair Gain From Trade objective is motivated by the so-called egalitarian rule in social choice theory (sometimes also called the max-min rule or the Rawlsian rule), where one favors the alternative that maximizes the minimum utility of the involved parties to promote fairness. We will add a paragraph about this bullet point in the revised version.

---

> > ### Comment · Reviewer_mLA5 · 2024-08-13
> >
> > Thank you to the authors for the detailed response. I have no additional questions.

---

### Author Rebuttal · Authors · 2024-08-06

We thank the four reviewers for the time spent reading our work and for sharing their comments.
We will update the submission in light of the feedback.
In particular, we will further highlight our technical contributions and the economic relevance of our results, we will provide additional discussions and clarifications in light of the comments and questions, and we will discuss potential extensions of our results. Additionally, we agree that subdividing the first section into "Introduction", "Related Work", and "Our Contributions" will improve readability.  The revised version will be structured as suggested.

Given the additional content page allowed, and the available time left, we are confident that we will successfully improve the submission with the requested changes.

We hope that our four separate responses have appropriately answered all questions and comments. We remain available should any further clarifications be needed.

---

### Decision · Program_Chairs · 2024-09-25

**Decision:**

Accept (poster)

**Comment:**

The paper studies an interesting and natural extension of the online bilateral trading model, where the goal is to maximize the minimum gain across the buyer and the seller, instead of the sum of their gains. The consensus is that the extension is well-motivated and has clear practical relevance. In addition the paper provides solid and comprehensive results on the achievable regret in various settings. Furthermore, the paper is well-written and well-structured. By tying well with the themes of fairness and online algorithms, the paper would be interesting to broader community.